# Moving towards malaria elimination in southern Mozambique: Cost and cost-effectiveness of mass drug administration combined with intensified malaria control

Laia Cirera[1]*, Beatriz Galatas[1,2], Sergi Alonso[3], Krijn Paaijmans[1,2,4], Miler Mamuquele[2], Helena Martí-Soler[1], Caterina Guinovart[1], Humberto Munguambe[2], Fabião Luis[2], Hoticha Nhantumbo[2], Júlia Montañà[1,2], Quique Bassat[1,2,5,6], Baltazar Candrinho[7], Regina Rabinovich[1,8], Eusebio Macete[2,9], Pedro Aide[2,9], Pedro Alonso[1,2©¤], Francisco Saúte[2©], Elisa Sicuri[1,10©]

1 ISGlobal, Hospital Clínic -Universitat de Barcelona, Barcelona, Spain, 2 Centro de Investigação em Saúde da Manhiça (CISM), Manhiça, Mozambique, 3 Centre for Primary Care and Public Health, Barts and The London School of Medicine & Dentistry, Queen Mary University of London, London, United Kingdom, 4 Center for Evolution and Medicine & The Biodesign Center for Immunotherapy, Vaccines and Virotherapy, School of Life Sciences, Arizona State University, Tempe, Arizona, United States of America, 5 Pediatric Infectious Diseases Unit, Pediatrics Department, Hospital Sant Joan de Déu (University of Barcelona), Barcelona, Spain, 6 ICREA, Barcelona, Spain, 7 National Malaria Control Program, Ministry of Health, Maputo, Mozambique, 8 Harvard T.H. Chan School of Public Health, Boston, Massachusetts, United States of America, 9 National Institute of Health, Ministry of Health, Maputo, Mozambique, 10 Department of Infectious Disease Epidemiology, Health Economics Group, School of Public Health, Imperial College London, London, United Kingdom

© These authors contributed equally to this work.
¤ Current address: Global Malaria Program, World Health Organization, Geneva, Switzerland
* laia.cirera@isglobal.org

**Data Availability Statement:** All relevant data are within the paper and its Supporting Information files.

## Abstract

### Background

As new combinations of interventions aiming at interrupting malaria transmission are under evaluation, understanding the associated economic costs and benefits is critical for decision-making. This study assessed the economic cost and cost-effectiveness of the Magude project, a malaria elimination initiative implemented in a district in southern Mozambique (i.e. Magude) between August 2015–June 2018. This project piloted a combination of two mass drug administration (MDA) rounds per year for two consecutive years, annual rounds of universal indoor residual spraying (IRS) and a strengthened surveillance and response system on the back of universal long-lasting insecticide treated net (LLIN) coverage and routine case management implemented by the National Malaria Control Program (NMCP). Although local transmission was not interrupted, the project achieved large reductions in the burden of malaria in the target district.

### Methods

We collected weekly economic data, estimated costs from the project implementer perspective and assessed the incremental cost-effectiveness ratio (ICER) associated with the

**Funding:** We acknowledge support from the Spanish Ministry of Science, Innovation and Universities through the "Centro de Excelencia Severo Ochoa 2019-2023" Program (CEX2018-000806-S), and support from the Generalitat de Catalunya through the CERCA Program. CISM is supported by the Government of Mozambique and the Spanish Agency for International Development (AECID). The Magude project (NCT02914145) was funded by the Bill and Melinda Gates Foundation and Obra Social "la Caixa" Partnership for the Elimination of Malaria in Southern Mozambique (OPP1115265). The funders had no role in study design, data collection and analysis, decision to publish, or preparation of the manuscript.

**Competing interests:** The authors have declared that no competing interests exist.

**Abbreviations:** AL, Artemether-lumefantrine; CE, Cost-effectiveness; CI, Confidence Interval; CISM, Centro de Investigação em Saúde de Manhiça; DALYs, Disability Adjusted Life Years; DDT, Dichlorodiphenyltrichloroethane; DHAp, Dihydroartemsinin-Piperaquine; GMEP, Global Malaria Eradication Programme; GTS, Global Technical Strategy; HF, Health Facility; ICER, Incremental Cost-effectiveness Ratio; IRS, Indoor Residual Spraying; LLIN, Long Lasting Insecticidal Net; MDA, Mass Drug Administration; MoH, Ministry of Health; NMCP, National Malaria Control Program; RDT, Rapid Diagnostic Test; rfMDA, Reactive Focal Mass Drug Administration; SSA, sub-Saharan Africa; WHO, World Health Organization.

Magude project as compared to routine malaria control activities, the counterfactual. We estimated disability-adjusted life years (DALYs) for malaria cases and deaths and assessed the variation of the ICER over time to capture the marginal costs and effectiveness associated with subsequent phases of project implementation. We used deterministic and probabilistic sensitivity analyses to account for uncertainty and built an alternative scenario by assuming the implementation of the interventions from a governmental perspective. Economic costs are provided in constant US$2015.

## Results

After three years, the Magude project averted a total of 3,171 DALYs at an incremental cost of $2.89 million and an average yearly cost of $20.7 per targeted person. At an average cost of $19.4 per person treated per MDA round, the social mobilization and distribution of door-to-door MDA contributed to 53% of overall resources employed, with personnel and logistics being the main cost drivers. The ICER improved over time as a result of decreasing costs and improved effectiveness. The overall ICER was $987 (CI95% 968–1,006) per DALY averted, which is below the standard cost-effectiveness (CE) threshold of $1,404/DALY averted, three times the gross domestic product (GDP) per capita of Mozambique, but above the threshold of interventions considered highly cost-effective (one time the GDP per capita or $468/DALY averted) and above the recently suggested thresholds based on the health opportunity cost ($537 purchasing power parity/ DALY averted). A significantly lower ICER was obtained in the implementation scenario from a governmental perspective ($441/ DALY averted).

## Conclusion

Despite the initial high costs and volume of resources associated with its implementation, MDA in combination with other existing malaria control interventions, can be a cost-effective strategy to drastically reduce transmission in areas of low to moderate transmission in sub-Saharan Africa. However, further studies are needed to understand the capacity of the health system and financial affordability to scale up such strategies at regional or national level.

## Introduction

An infectious disease is considered eliminated in a specific geographical area when its local transmission is interrupted and maintained at zero [1, 2]. In the long term, the economic rationale for eliminating infectious diseases is apparent: if elimination is achieved, a high cost-effectiveness and high benefit-cost ratios compared to continued disease control are guaranteed and constitute the so-called dividend (i.e. the profit of a financial investment) [3]. Long-term benefits are the result of the improvement of both health and non-health outcomes, such as management and treatment cost-savings due to reduced cases and deaths, improved productivity and labour supply, increased educational attainment [4] and literacy rates [5] and higher lifetime earnings and occupation rates [6], all contributing to economic growth and socioeconomic development.

While targeting elimination in the long term is desirable—and achievable for several infectious diseases—, the extremely high costs, combined with the uncertainty and associated risk of failure, cast doubts on short-term feasibility and efficiency of elimination strategies [7–10]. Equity concerns also factor into decision-making, as during the initial stages of elimination the less challenging and easy to reach areas and/or groups may be targeted, often leaving the most vulnerable and poor population aside [11]. Other challenges stem from the fact that disease elimination is a global public good, characterized by the non-excludability and non-rivalry attributes in consumption [12]. The 'global public good' concept implies that governments need to coordinate financial mechanisms and boost cooperation at the regional level in order to achieve elimination as a common goal. Importantly, these challenges should not only surface in the last mile preceding the actual achievement of elimination, but be tackled when decisions on control optimization or pre-elimination initiatives are being made.

All the issues above need to be addressed in a context of scarce financial resources and additional pressing public health priorities, where policymakers are faced with key economic questions such as: (1) are the costs associated with investments towards malaria elimination affordable and sustainable in a context of competing health challenges?; (2) is the increased effort associated with implementing interventions towards malaria elimination (with old and/or new interventions and/or strategies) economically justified in comparison with continuing with routine control interventions?

By comparing the incremental costs and health effects of elimination initiatives over time relative to alternative (often business-as-usual) scenarios, cost-effectiveness assessments provide essential instrumental evidence to answer such questions and inform policy-makers on how to best prioritize and allocate limited resources in the short term, while monitoring efficiency of activities towards elimination [13].

This debate becomes relevant in the context of malaria, where despite the progress made in the last decades, the burden of disease remains strikingly high, particularly in sub-Saharan Africa (SSA). As a result, the World Health Organization (WHO) Global Technical Strategy for Malaria 2016–2030 (GTS) has urged for the generation of evidence on effective strategies—using available tools—to accelerate progress towards elimination [14].

The combination of mass drug administration (MDA) for malaria, consisting of door-to-door administration of antimalarial treatment to every member living in a defined geographical area on the back of existing prevention and treatment tools, has increasingly received attention as a promising strategy to rapidly reduce transmission in low to moderate transmission settings [15]. Although MDA was part of control and elimination strategies during the Global Malaria Eradication Programme (GMEP) in the 1950s and –60s, evidence on its effectiveness is limited [16, 17]. Recent studies conducted in Comoros islands, Zambia and South East Asia [18–20] have reported that MDA using dihydroartemisinin piperaquine (DHAp) is effective in reducing—although not interrupting—*P. falciparum* malaria transmission to unprecedented low levels.

More recently, the Magude project has assessed the feasibility of achieving malaria elimination in an endemic district in southern Mozambique. Following the GTS recommendations [14], the project combined an optimized package of existing interventions, including a strengthened surveillance system, case management, intensified vector control with universal long-lasting insecticide treated net (LLIN) distribution and universal (i.e. targeted to all households in the district) indoor residual spraying (IRS), and mass drug administration (MDA) [21]. The project was based on direct implementation of malaria interventions and was managed on a learning-by-doing basis, with resources for MDA delivery adjusted over time based on experience accumulation. Effectiveness results of the Magude project align with existing

evidence [19, 20], indicating that the package of interventions did not interrupt malaria transmission but drastically reduced malaria prevalence and incidence [22].

To date, the debate has largely focused on the impact of MDA on malaria prevalence, but its cost-effectiveness is poorly understood due to the lack of accurate data on costs and resources for MDA campaigns. Scarce short and long-term information is available as either aggregate financial costs from past GMEP elimination experiences, or for very specific settings, such as islands or emergency scenarios [23], which limits its use in current programme planning in countries approaching elimination [10].

As a result, knowing the short-term costs and benefits associated with strategies involving MDA becomes critical to guide policy-makers in prioritizing and sustaining resources while transitioning from malaria control to malaria elimination. Mozambique is one of the highest malaria burden and weakest link countries in southern Africa, contributing to cross-border transmission and impeding the achievement of a malaria-free status in neighbouring countries [24]. Insights on efficient resource allocation become essential for accelerating progress towards malaria elimination at the national and regional level.

In this study, the economic resources and cost-effectiveness of the intervention package implemented in the Magude project are compared with those associated to routine prevention, diagnosis and treatment interventions under routine malaria control (i.e. annual rounds of focal IRS, LLIN distribution and standard case management).

## Methods

### Study site

Magude district is a rural district in Maputo province, southern Mozambique, with 48,448 identified individuals and 10,965 households, according to a baseline census from 2015. The district has year-round malaria transmission, with seasonal peaks in the rainy season (November–April). Further epidemiological and socio-demographic characteristics of the district have been described elsewhere [22, 25].

The package of interventions deployed at the district level under the Magude project consisted of: a) a strengthened epidemiological surveillance reporting systems established in the district since January 2015; b) annual rounds of universal IRS using DDT and/or pirimi-phos-methyl (Actellic,®) between August–October of 2015, and between September–November of 2016, 2017 and 2018; c) two yearly rounds of MDA during two consecutive years, deployed in November 2015 (MDA1), January–February 2016 (MDA2), December 2016 (MDA3) and January–February 2017 (MDA 4) and d) community engagement for MDA to maximize the acceptance of the intervention followed by e) an active surveillance system with focal MDA in the index case household, or reactive focal MDA (rfMDA), starting on June 2017 (Fig 1).

The impact of the interventions was estimated by conducting a before-after study and employing interrupted time series analysis on passively detected weekly malaria cases (by RDT or microscopy) at the health facilities or by community health workers. MDA coverages varied between 58–72% across the four rounds, with children under-five years receiving a higher protection (>70%) in comparison with population older than five (50–70%). Within three years (August 2015–June 2018), parasite prevalence decreased by 86% and case incidence fell from 195 to 67 cases per 1,000. As a result, an estimated 76.7% of expected cases were averted (38,369 cases averted of 50,005 expected cases had the intervention not taken place) between August 2015 and June 2018. Further details on interventions coverages and effectiveness measurement have been reported elsewhere [22].

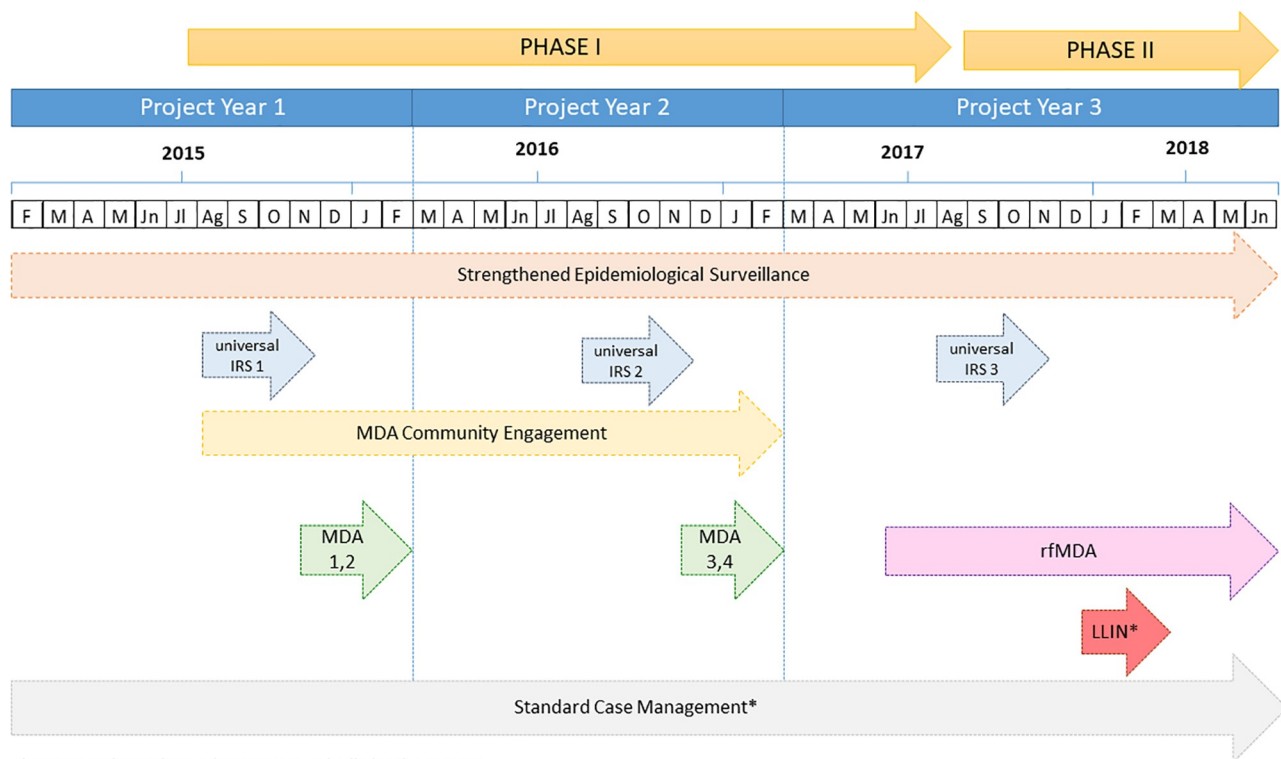

**Fig 1. Chronogram of activities and interventions of the Magude project, 2015–2018.** Chronogram of the main malaria elimination interventions implemented in the study district between January 2015 and June 2018, by project year and interventions phase (I and II). MD, mass drug administration; IRS, indoor residual spraying; rfMDA, reactive focal mass drug administration; LLIN, long-lasting insecticidal nets.

## Study design

We used an incremental approach from the implementer's perspective and compared the costs of the interventions implemented in phase I (between August 2015–2017) and phase II (September 2017–June 2018) of the project (Fig 1) with the costs of routine malaria control (i.e. counterfactual scenario).

The counterfactual scenario included routine malaria vector control activities implemented by the government, which include annual rounds of focal IRS (i.e. spraying targeted to high-burden areas in the district), universal LLIN distributions during national campaigns every three years, and prompt diagnosis and provision of treatment with efficacious anti-malarial drugs. According to national guidelines [26], standard case management is delivered at the health facilities (HF), where malaria testing is performed with rapid diagnostic tests (RDTs) available at all levels, first-line treatment is artemether-lumefantrine (AL) for uncomplicated malaria cases and injectable artesunate for all severe cases.

The interventions planned programmatically by the National Malaria Control Programme (NMCP) that took place in the district simultaneously, including standard case management as well as a national mass distribution of long-lasting insecticidal treated nets (LLIN), were also computed as part of the overall package of interventions within the project (Fig 1).

## Costs and DALYs

We collected weekly data on the economic resources employed by the project since its inception in 2015 and developed an ingredient-based costing (micro-costing). Regular meetings

were held with each area's responsible to list the quantity of resources—economic and financial—utilized in each activity.

Unit costs of purchased resources, disaggregated by commodity and importation price, and in-country delivery costs, were obtained from the procurement department. For non-financial items (i.e. resources used by the project in implementing the interventions that did not involve a financial transaction, such as donated goods or volunteers time), we used data from published literature and country evidence (S1 Table). For outsourced activities, such as universal IRS (implemented by GoodBye Malaria to all households in the district), we revised the executed expenditures and organized costing items to fit with our approach.

Costs were depreciated, annualized, inflated, discounted and/or allocated as shared resources according to methodological guidelines [27–29] and expressed in constant 2015 US$ using average yearly exchange and inflation rates [30, 31] (S1 and S2 Tables). Our analysis did not consider costs unrelated to the operational aspects of running the project, such as research costs, and merely approximates the costs incurred by the government if the project was to be scaled-up to other areas (see S1 Text for details on costing formulas employed).

Cost data associated with routine malaria vector control interventions as part of the counterfactual scenario were gathered from the Global Fund's Price and Quality Reporting database (for LLIN,) and the President's Malaria Initiative country evidence (for IRS) [32]. IRS coverage rates (52.2%) registered in the Magude demographic census in 2015 were used to estimate the costs of routine IRS [25].

Standard case management costs associated with the Magude project were calculated considering the observed malaria cases across time. In addition, case management costs under the counterfactual scenario were based on the estimated cases had the intervention not taken place. Treatment unit costs included recurrent costs such as personnel, drugs and supplies costs incurred in an outpatient visit. For malaria admission, injectable artesunate treatment and admission costs (assuming an average of 5 days based on expert consultation) were also considered. Treatment unit costs (for outpatients and inpatients) were gathered from a previous study carried out in the district and updated from 2007 to 2015 figures using an inflation rate correction factor [33]. Incremental costs are expressed in net terms, as they considered cost-savings due to treating fewer malaria cases under the Magude project scenario.

We translated the estimated number of malaria cases averted associated with the project into DALYs averted. Based on evidence from a neighbouring district hospital, we used the fraction of outpatient visits for malaria that required hospitalization and assumed it to be equivalent to the percentage of malaria cases that derive into severe malaria (even though not all inpatients might have been diagnosed with severe malaria). Evidence on inpatient case fatality rate from the district hospital was used and assumed to reflect the fatality rate of severe malaria cases (S1 Table).

DALYs averted were estimated by multiplying the number of DALYs lost from malaria morbidity (severe and non-severe malaria cases) and mortality times the effectiveness of the project on reducing malaria cases and deaths, respectively [28]. DALYs were discounted and calculated according to conventional approaches [28] (see S1 Table for key parameters and sources used in DALYs calculation). Aligned with recent consensus among experts [34], DALYs have not been aged-weighted in the analysis.

## Data analysis

The ICER was calculated by dividing the net incremental costs associated with the Magude project by the DALYs averted by the project, when compared to routine malaria control

activities under the counterfactual scenario:

$$ICER = [(Cost\ Magude\ project + Cost\ case\ management\ Magude\ project) - (Cost\ routine$$
$$control + Cost\ case\ management\ routine\ control)]/[DALYs\ associated\ Magude\ project -$$
$$DALYs\ associated\ routine\ control]$$

To capture potential economies of scale and scope, we estimated the ICER at three different timepoints: i) by end year 1, after the deployment of MDA1 and MDA2 (August 2015–February 2016); ii) by end year 2, after the deployment MDA3 and MDA4 (August 2015–February 2017) and iii) by end year 3, one year after the discontinuation of MDA (August 2015–June 2018).

We varied specific parameters to assess their contribution to overall outcomes, understand key costing drivers and take into consideration univariate uncertainty (S3 Table). We built a basic alternative scenario in which several parameters were adjusted in order to estimate the costs of the project if it was implemented by the government. The Magude project implemented from a governmental perspective consisted in adjusting wages and per diems to those paid by the MoH—according to the corresponding health professional category [35]—and in considering the use of already existing capital goods within the public health system (i.e. vehicles, warehouses and health structures) and applying the corresponding depreciation rate, instead of being computed as a purchase or rental. A deterministic threshold sensitivity analysis was performed to estimate the minimum number of cases averted for the Magude project to be considered cost-effective.

Joint parameter uncertainty was considered by expressing all model inputs as probabilistic according to appropriate distribution functions [36], with assumed uncertainty range of 20% applied to each parameter point estimate, except for parameters for which specific evidence on uncertainty ranges was available (S1 Table), and conducted Monte Carlo simulations (1,000 iterations).

Probabilistic results were plotted in a cost-effectiveness plane. To define the Magude project as cost-effective in comparison with routine malaria control (counterfactual scenario), we primarily used standard cost-effectiveness thresholds, based on thresholds of one (highly cost-effective) to three times (cost-effective) the Mozambican gross domestic product (GDP) per capita [37], averaged across the period of study. We also graphically represented our results as acceptability curves, which show the probability of the project of being cost-effective for different willingness to pay values, and compared our ICER results with alternative thresholds based on the health opportunity cost [38].

A long-term scenario was built by extending both the Magude project and counterfactual scenario costs to 2030. In this modelling exercise, we assumed that the malaria incidence levels achieved by the project would be maintained with continued vector control and rfMDA as implemented during the third year. Malaria routine activities and incidence in the counterfactual scenario were assumed to remain stable over time. Malaria incidence figures were adjusted for population growth rates [39] (see S2 Text for details).

## Results

The economic cost of the Magude project over 3 years was $4.33 million, with the four rounds of MDA being the most resource intense activity, accounting for 53% of overall resources. With the inclusion of case management costs, at an outpatient and inpatient cost of $1.7 and $175, respectively, per malaria episode treated, total economic project costs amounted to $4.83 million (Table 1).

**Table 1. Costs of the Magude project vs counterfactual scenario (routine malaria control).**

| Activity | The Magude project | | Counterfactual Scenario | Difference | Comments |
|---|---|---|---|---|---|
| | Mean (US$) | Contr. (%) | Mean (US$) | Mean (US$) | |
| Epidemiological surveillance | 326,260 | 8% | - | 326,260 | |
| Mass Drug Administration | 2,297,626 | 53% | - | 2,297,626 | |
| Community engagement (for MDA) | 224,981 | 5% | - | 224,981 | |
| Universal IRS | 1,243,128 | 29% | - | 1,243,128 | IRS implemented by the Magude project, targeted to all households in the district. It achieved operational coverages higher than 90% [23] |
| Focal IRS | - | - | 473,836 | -473,836 | IRS implemented by the NMCP, targeted to households in high-burden areas of the district. It achieved coverage rates of 52.2% [33]. PMI reference unit costs [32] |
| rfMDA | 186,746 | 4.31% | - | 186,746 | |
| Universal LLIN distribution * | 54,168 | 1.25% | 54,168 | 0 | Mass LLIN distribution planned programmatically by the NMCP in December 2017 |
| **Sub-total** | **4,332,909** | | **528,005** | **3,804,904** | |
| Case management costs* | | | | | |
| Outpatient | 35,761 | | 101,444 | -65,684 | Cost savings due to reduced burden of disease under the Magude Project |
| Inpatient | 462,308 | | 1,311,456 | -849,149 | |
| **Total costs (net)** | **4,830,977** | | **1,940,905** | **2,890,072** | |
| malaria cases | 20,889 | | 59,257 | **38,369** | Averted malaria cases Magude project |
| DALYs | 1,726 | | 4,897 | **3,171** | Averted DALYs Magude project |
| **ICER (deterministic)** | | | | **912** | USD$ / DALY averted |

MDA, mass drug administration; IRS, indoor residual spraying; rfMDA, reactive focal mass drug administration; LLIN, long lasting insecticidal nets; NMCP, national malaria control programme; Contr, contribution; Ref, reference. Costs in constant US$ 2015.

* Interventions planned programmatically by the NMCP

Main costing drivers across activities were personnel (39%) and transportation (22%) resources, followed by malaria drugs and other supplies (19%) (S4 Table). Ten percent of employed resources included non-financial costs. Non-financial costs were especially present in activities implemented at the health facilities level, such as those related to strengthening the surveillance system, given the use of existing resources within the public health system. Aligned with previous experiences from other door-to-door MDA interventions [23], the main cost drivers of the community-wide drug administration—with an average cost of $19.4 per person treated/round—were personnel and transportation. However, MDA costs decreased by approximately 50% every two rounds, as fewer resources (especially those related to personal and transportation) were used in a shorter time span, leading to a drop from $26 for rounds 1 and 2 to $13 for rounds 3 and 4 per person treated and per round (S5 Table).

When compared to the counterfactual scenario, the Magude project, at a net incremental cost of $2.89 million, averted a total 38,369 malaria cases and 3,171 DALYs, leading to a deterministic ICER of $912 per DALY averted (Table 1). In the base case, the ICER was lower than the standard cost-effectiveness threshold ($1,404/DALY averted), but higher than the threshold of interventions considered highly cost-effective ($468/DALY averted) [28]. Results showed that the project would no longer be cost-effective with less than 24,936 malaria cases averted, equivalent to 2,061 DALYs averted. These figures represent a 35% decrease in effectiveness.

The one-way sensitivity analysis reflected that the ICER is extremely sensitive to the malaria case fatality rate (CFR) (S3 Table; Fig 2a). Changing MDA operational aspects such as field-workers' efficiency (in terms of increased number of houses visited per day and per team) and assuming a constant efficacy rate, also resulted in significantly different MDA cost per capita and ICER results. On the other hand, results showed little variation to changes in operational or costing parameters related to IRS implementation (S2 Table; Fig 2b and 2c).

When evaluated over time (by the end of project year 1, year 2 and year 3), the ICER showed a decreasing trend, reflecting decreased marginal costs but also increased marginal effectiveness during the project's implementation timeline (S1 and S2 Figs). Adjusting model parameters to reflect the Magude project implementation costs from a governmental perspective (i.e. costs of the same activities when implemented by the NMCP) significantly reduced the costs (by 32%), leading to an ICER below the highly cost-effectiveness threshold ($441/ DALY averted) (S6 Table and S1 Fig).

Table 2 presents the results from the probabilistic analysis, with an ICER of $987 (CI95% 968–1,006) per DALY averted (incremental cost of $2.89 million [2.86–2.90] and 3,167 incremental DALYs averted [3,111–3,223]), or an equivalent $75 per malaria case averted.

The cost-effectiveness plane plots probabilistic results and suggests that even when accounting for parameters' uncertainty, the project could remain cost-effective by end year 3 (June 2018), with 91% of the simulation points concentrated below the cost-effectiveness threshold ($1,404 per DALY averted) (Fig 3). The acceptability curves complement this information and show the probability of the project being cost-effective for a range of different willingness to pay values (to be determined based on governmental preferences, availability of resources, etc.) per DALY averted (Fig 4) or malaria case averted (S3 Fig).

Finally, basic long-term costing suggests that, under the assumption that the gains achieved by the Magude project interventions could be sustained through focalized approaches combined with continued vector control and standard case management, as in the third year of the project [40], the Magude project would potentially become a cost-saving strategy by 2030, with financial benefits resulting from treating fewer malaria cases exceeding the initial project costs (S2 Text and S4 Fig).

## Discussion

This study shows that the economic cost of the Magude project was substantially higher than the routine malaria control activities that would have otherwise taken place in the district. In spite of higher absolute costs, the project was cost-effective by the end of year 3, with an ICER of $987 (CI95% $968–1,006) per DALY averted, a value below the conventional cost-effectiveness threshold of three times the GDP per capita (S1 Fig). We estimate that the project would have still remained cost-effective if achieving at least 65% of the effectiveness (i.e. number of cases averted) observed. This suggests that the mix of interventions delivered through the Magude project would potentially remain cost-effective if implemented in less favourable environmental and socioeconomic contexts.

The ICER decreased over time, from $7,414 to $987 per DALY averted between the first and third year of interventions. This contradicts the outcome of previous studies showing an increasing ratio when optimized malaria control interventions are added [41, 42]. The ICER in this project decreased as a result of decreasing project costs (at the numerator) and increasing project effectiveness (at the denominator). Taking into consideration that MDA coverage remained relatively constant but adherence to the drug regime decreased between rounds (S7 Table), the effectiveness trend may reflect two aspects: (1) the cumulative increasing health effects of the package of interventions implemented and (2) the high effectiveness of continued

(a)

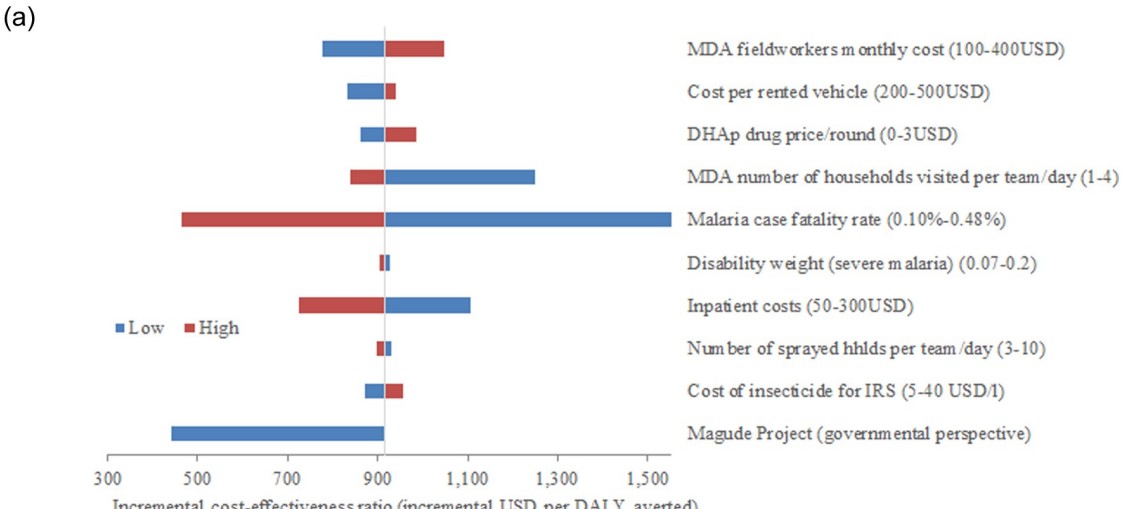

(b)

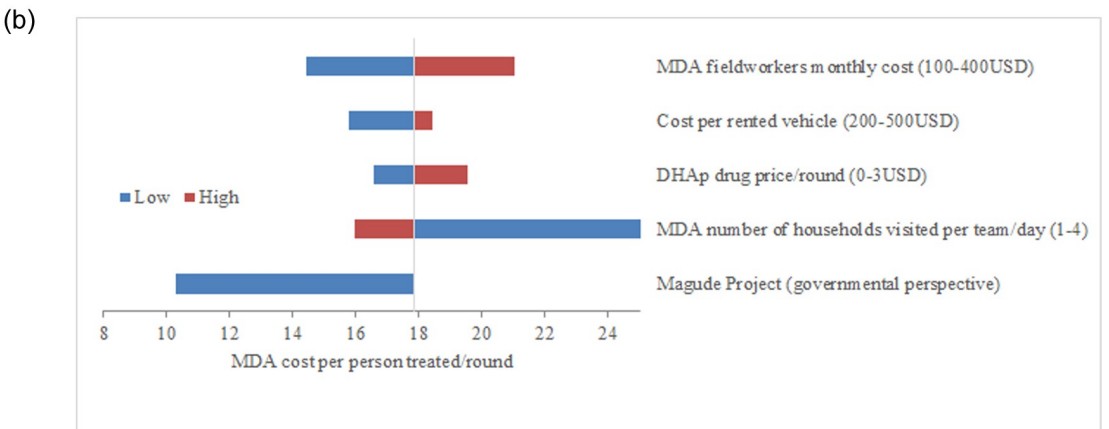

(c)

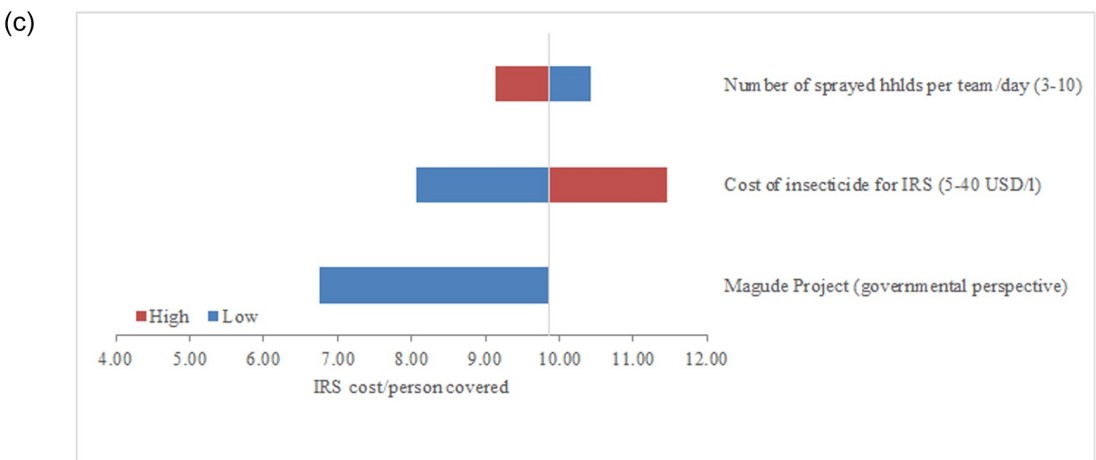

**Fig 2.** (a, b, c). Tornado diagram, deterministic sensitivity analysis. The values in the parentheses stand for the lower and higher range over which the parameter was varied. The vertical line represents the baseline value of the outcome being analysed: ICER in Fig 2a, MDA cost per person treated/round in Fig 2b and universal IRS cost/person covered in Fig 2c. The blue bars show the direction and magnitude of change in the outcome of interest, when the input variable is set to its lower range and the red bars show the direction and magnitude of change when the input variable is set to its higher range. See S1 Table for further details. DALYs,

disability-adjusted life years; MDA, mass drug administration; DHAp, dihydroartemisinin piperaquine; IRS, indoor residual spraying.

**Table 2.  Monte Carlo simulation results of the Magude project.**

| | Differences | | | | | | |
|---|---|---|---|---|---|---|---|
| | Incr Cost (2015 US$) | DALYs averted | | Cases averted | ICER (2015 US$) | | |
| | Range | Range | Mean | | Range | Mean | Median |
| Magude project end Y1 | (1,9181,21–1,933,915) | (277–287) | 282 | 3,417 | (7,272–7,556) | 7,414 | 6,979 |
| Magude project end Y2 | (2,855,967–2,875,219) | (1,108–1,148) | 1,128 | 13,668 | (2,707–2,811) | 2,759 | 2,590 |
| Magude project end Y3 | (2,862,766–2,896,358) | (3,111–3,223) | 3,167 | 38,374 | **(968–1,006)** | **987** | 933 |

intensified vector control and rfMDA in maintaining the gains achieved. The increase in effectiveness could also be driven by environmental factors, given that 2015–16 was an unusual dry malaria season, whereas 2016–17 was characterized by particularly high rainfall, which resulted into very low and exceptionally high malaria incidence levels, respectively.

The observed downward cost trend throughout project implementation, on the other hand, can be explained by a reduction in annualised economic MDA costs from round to round

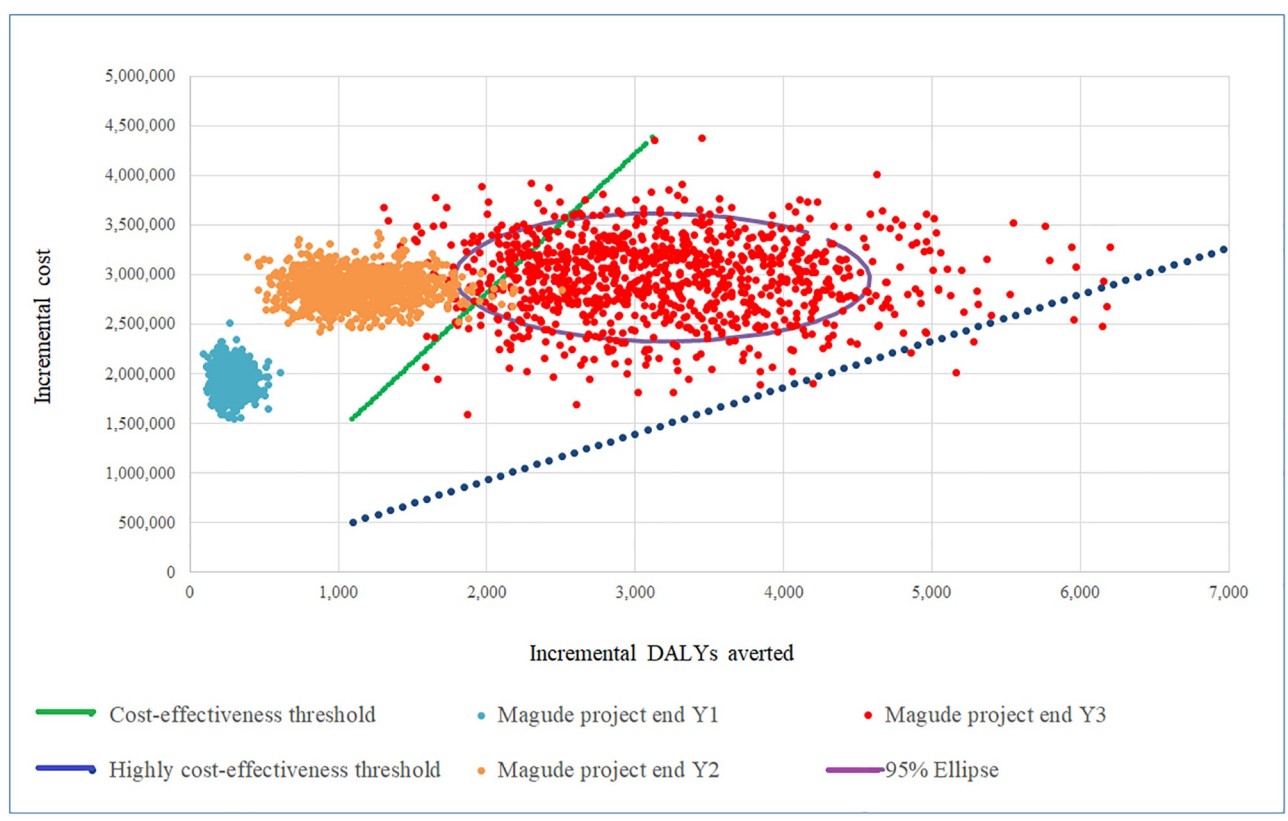

**Fig 3. Cost-effectiveness plane.** This figure plots the incremental costs (Y axis) and the averted DALYs (X axis) from the Magude project after its first year (light blue dots), after its second year (orange dots) and after its third year (red dots). The circle represents the 95% ellipse (the 95% credible interval); the blue dashed line represents the standard high cost-effectiveness threshold equal to one time the gross domestic product per capita ($468 per DALY averted) and the green line represents the standard cost-effectiveness threshold equal to three times the gross domestic product per capita ($1,404 per DALY averted). DALYs, disability-adjusted life years.

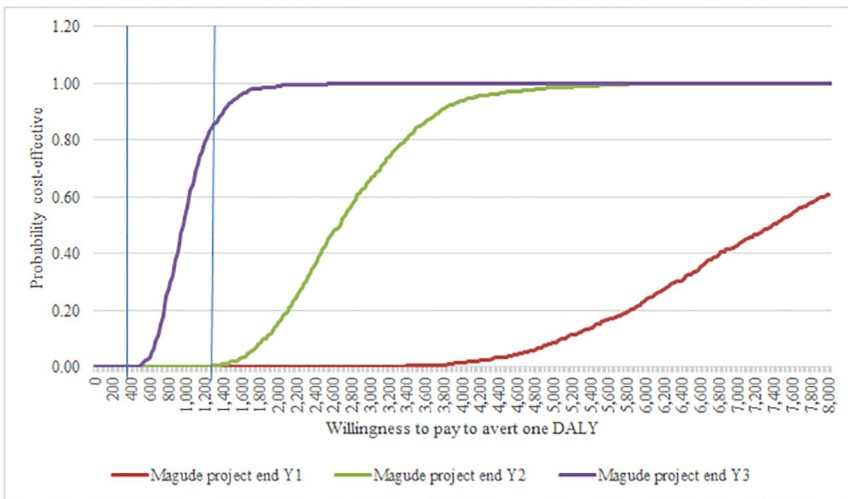

**Fig 4. Cost-effectiveness acceptability curve.** The acceptability curves show the probability that the Magude project is cost-effective (compared to continuing with routine malaria control) across time (by end year 1, year 2 and year 3) for different levels of willingness to pay to avert one DALY (X axis). The vertical lines represent different WTP that can be applied to Mozambique: US$468 per DALY averted (standard threshold of highly cost-effective intervention; and US $1,404 per DALY averted (standard threshold of cost-effective intervention). DALYs, disability-adjusted life years.

(from $26 to $13 per person treated) due to the accumulation of know-how (a similar operational coverage was achieved in rounds 3 and 4, employing nearly half of the workforce from rounds 1 and 2), the reduced training needs and an improvement in resource planning and organization [43], together with reduced case management and treatment costs. If a longer time span is considered, the ICER better reflects the potential cumulative cost-savings that accrue from reducing the burden of disease (S4 Fig).

Direct comparison of our costs estimates with other door-to-door MDA unit costs would not be appropriate, given that available evidence refers to very diverse contexts, epidemiological settings and MDA specific purposes. However, available estimates—ranging from $1.22 in Sierra Leone to $14.13 in Comoros Island [23]—also reflect a large share of personnel and transportation costs used for MDA. In addition, whilst a decreasing marginal unit cost over time was observed, these estimates reflect the costs associated with the intensification of malaria control efforts, and not necessarily the costs of reaching the last mile of disease elimination. Several studies estimating the costs associated with modelled elimination (or even eradication) found the cost of averting a marginal case to exponentially increase when approaching the last mile [7, 44]. We can speculate that, should such efforts continue in Magude, the unit cost would rise until elimination is achieved.

The use of conventional cost-effectiveness thresholds in economic assessments has been subject to considerable debate. Recent empirical estimates of country-specific opportunity cost suggest significantly lower thresholds for Mozambique ($ 537 purchasing power parity per DALY averted or $ 294 unadjusted) [38]. If these estimates were considered in this study, the project's cost-effectiveness would be debatable. More importantly, cost-effectiveness thresholds are only simplified indications on what may constitute a poor, good or very good value for money, and these should be used alongside other criteria that reflect a country's affordability and willingness to pay, as well as other dimensions instrumental in the decision-making process (e.g. equity). The acceptability curves provide a broader spectrum for results interpretation within a context of uncertainty and beyond pre-determined thresholds. If the government

sought a 95% probability of the Magude project being cost-effective, it should be willing to pay at least $1,500 per DALY averted.

However, one should be careful with the extrapolation of results to other malaria settings. First, some specific activities that have taken place in the Magude project are not necessarily an integral part of standard core malaria elimination interventions. Such activities may have influenced the project's effectiveness, by steering decisions on key interventions implementation. For example: a) the demographic census facilitated the identification of households and residents in the area, enhancing MDA and IRS operational coverage and b) the studies on insecticide resistance of malaria vectors guided the selection of appropriate insecticides for effective IRS. The cost imputation of these activities to the project would not only be difficult, as those refer to research activities implemented for other purposes (i.e. identifying baseline demographic, epidemiological and entomological indicators from new studies), but also inappropriate, given they are not representative of the activities that would be implemented if the project was replicated at a larger scale. Nonetheless, their inclusion would not have altered our findings significantly, as the associated costs are relatively small. Second, the project was implemented by external organizations, which means that associated resources and costs do not reflect those occurring under an implementation in programmatic mode. To illustrate this, by adjusting salaries to governmental norms and assuming the use of already existing public and governmental infrastructures, the costs would decrease by 32%. If this program were to be equally effective, this would translate to an ICER of $441 per DALY averted.

In addition, the presence of potential economies of scope may improve the ICER even further. On example are community-based health interventions run by the government as part of the NMCP or other programmes [45]. MDA programs are operational in Mozambique for the control and elimination of lymphatic filariasis, schistosomiasis and soil-transmitted helminths since 2011. As a result, the costs per person treated have diminished by 60% and compliance rates have improved since initiation of the program [46]. As MDA interventions are scaled-up, economies of scale can be expected as well [47]. Recent estimates for neglected tropical diseases point to a cost of less than $0.5 per person when more than 100,000 people are treated [48]. This figure may be achievable for MDA for malaria as well [47].

While other economic questions related to equity, scalability, sustainability and financial affordability associated with moving from control to pre-elimination remain unanswered, this study offers solid evidence on the economic rationale for prioritizing resources on innovative strategies that accelerate the progress towards malaria elimination. The micro-costing approach presented here also provides essential evidence on key inputs for costing extrapolation and scenario development in other settings. Despite the initial high costs and volume of resources associated with its implementation, MDA in combination with existing malaria control interventions appears a potentially cost-effective strategy to accelerate towards malaria elimination in low to moderate transmission settings in SSA.

## Supporting information

**S1 Fig. ICER evolution ("direct evidence Magude project" and "governmental perspective").**
(DOCX)

**S2 Fig. Cumulative malaria cases averted across time (2015–2018).**
(DOCX)

**S3 Fig. Cost-effectiveness acceptability curve per malaria case averted.**
(DOCX)

**S4 Fig. Cumulative costs across time, 2015–2030 (US$ million).**
(DOCX)

**S1 Table. Input variables and probabilistic distribution for cost-effectiveness analysis.**
(DOCX)

**S2 Table. Allocation of Magude project shared resources.**
(DOCX)

**S3 Table. Deterministic sensitivity analysis, parameters inputs and results implications.**
(DOCX)

**S4 Table. Economic costs per budget category and activity (2015–2018).**
(DOCX)

**S5 Table. MDA costs per budget category and round.**
(DOCX)

**S6 Table. The Magude project costs ("direct evidence Magude project" vs. "governmental perspective").**
(DOCX)

**S7 Table. MDA coverage and adherence rates across rounds.**
(DOCX)

**S1 Text. Costing formulas.**
(DOCX)

**S2 Text. Costs projection assumptions (2015–2030).**
(DOCX)

## Acknowledgments

We thank the community of Magude and the district authorities for allowing this project to take place. We also thank the Ministry of Health and the National Malaria Control Program for their contribution to the successful implementation of the project. We extend our gratitude to the Global Malaria Program at WHO, for their guidance and technical support. Special thanks go to Dr Edith Patouillard, for her relevant comments and contributions in the revision of the final manuscript.

## Author Contributions

**Conceptualization:** Laia Cirera, Pedro Alonso, Francisco Saúte, Elisa Sicuri.

**Data curation:** Laia Cirera, Beatriz Galatas, Krijn Paaijmans, Miler Mamuquele, Helena Martí-Soler, Caterina Guinovart, Humberto Munguambe, Fabião Luis, Hoticha Nhantumbo.

**Formal analysis:** Laia Cirera, Elisa Sicuri.

**Methodology:** Laia Cirera, Beatriz Galatas, Sergi Alonso, Helena Martí-Soler, Elisa Sicuri.

**Writing – original draft:** Laia Cirera.

**Writing – review & editing:** Laia Cirera, Beatriz Galatas, Sergi Alonso, Krijn Paaijmans, Helena Martí-Soler, Caterina Guinovart, Júlia Montañà, Quique Bassat, Baltazar

Candrinho, Regina Rabinovich, Eusebio Macete, Pedro Aide, Pedro Alonso, Francisco Saúte, Elisa Sicuri.

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
