## [Decision Letter · Decision Letter 0]

5 May 2020

PONE-D-20-09618

Moving towards malaria elimination in Southern Mozambique: cost and cost-effectiveness of mass drug administration combined with intensified malaria control

PLOS ONE

Dear Crivillé,

Thank you for submitting your manuscript to PLoS ONE. After careful consideration, we felt that your manuscript requires revision, following which it can possibly be reconsidered. Although your manuscript was of interest to the reviewer, a significant amount of issues should be clarified including population size, local malaria endemicity, and  the applicability of the findings according to WHO’s guidelines.    Finally, the MS should be submitted to a copy-editing process otherwise the readability of the MS is compromised. For your guidance, a copy of the reviewers' comments was included below 

We would appreciate receiving your revised manuscript by  July 20. To enhance the reproducibility of your results, we recommend that if applicable you deposit your laboratory protocols in protocols.io, where a protocol can be assigned its own identifier (DOI) such that it can be cited independently in the future. For instructions see: http://journals.plos.org/plosone/s/submission-guidelines#loc-laboratory-protocols

We look forward to receiving your revised manuscript.

Kind regards,

Luzia Helena Carvalho, Ph.D.

Academic Editor

PLOS ONE

Journal Requirements:

1. Thank you for including your ocmpeting interests statement; "The authors have declared that no competing interests exist."

We note that one or more of the authors are employed by a commercial company:ICREA, Pg. Lluís Companys

2. Thank you also for providing the following funding information within your acknowledgements section; "CISM is supported by the Government of Mozambique and the Spanish Agency for International Development (AECID). The *Magude project* (NCT02914145) was funded by the Bill and Melinda Gates Foundation and Obra Social “la Caixa” Partnership for the Elimination of Malaria in Southern Mozambique (OPP1115265)."

Reviewers' comments:

Reviewer's Responses to Questions

**Comments to the Author**

1. Is the manuscript technically sound, and do the data support the conclusions?

Reviewer #1: Yes

Reviewer #2: Yes

Reviewer #3: Yes

2. Has the statistical analysis been performed appropriately and rigorously? 

Reviewer #1: Yes

Reviewer #2: Yes

Reviewer #3: Yes

3. Have the authors made all data underlying the findings in their manuscript fully available?

Reviewer #1: Yes

Reviewer #2: Yes

Reviewer #3: Yes

4. Is the manuscript presented in an intelligible fashion and written in standard English?

Reviewer #1: No

Reviewer #2: Yes

Reviewer #3: Yes

5. Review Comments to the Author

Reviewer #1: General comments:

This study evaluates the cost-effectiveness of a pilot malaria elimination program in Mozambique called the Magude project. It demonstrates, like other elimination studies, that drastic disease reductions are achieved during active efforts; but unfortunately, these efforts tend to fail in eliminating local transmission and therefore disease burden climbs again once MDA is stopped. Correspondingly, the ICER of the project improved with each MDA round but ultimately did not achieve a highly cost-effective threshold by the end of the project.

Since I am not a health economist expert, but have a great deal of experience with malaria elimination, the methodology of the study design and analysis appear sound and the figures are clear. However, my concerns are two-fold:

1) the overall manuscript is not clearly written or punctuated such that it was difficult to read

2) the content may be more appropriate to a malaria-specific journal and less so to the readership of PLOS One.

Undoubtedly, as with all malaria elimination programs it was a great deal of important work and should be published somewhere. I therefore recommend major revision.

Specific points:

Grammar, punctuation and sentence restructuring are required throughout manuscript but here are some examples:

- The English is a bit stilted, for example “in the long-term” is not a correct expression and is used multiple times (Lines 24, 31). Long-term is an adjective not a noun. Also elimination is maintained “at zero” not “to zero” (line 24) and lines 43-49 are confusing and not clear. This and the next paragraph have only single sentences in them. There are many run-on sentences in intro that impact clarity. Would recommend a thorough read and edit by a native English speaker.

- Lines 64-67 should be in results or discussion, not intro

- Lines 96-97 need to use different distinguishers for MDA rounds and time frames - too many short hyphens. Figure 1 is nice and clear.

- Inappropriate use of hyphen versus em dash throughout (or inconsistent - see lines 199-200)

- Lines 134-135 --- poorly written and punctuated “Based on the observed malaria cases and the estimated cases expected had the intervention not taken place - the counterfactual-, respectively, we estimated resources…”

- Line 269 - should be em dashes

- Line 318 - “Mozambique stands among the weakest link countries” - this does not make sense to use stand in this sentence - Mozambique IS one of the weakest links in southern Africa

Abstract

- Last sentence of background should be in results

- Inappropriate use of short hyphens throughout

Introduction/Methods

- This is called a proof of concept pilot study in abstract but I do not see any Magude Project references or actual descriptions of the study interventions itself aside from Figure 1 and corresponding test. It would be nice to know more to set the stage - e.g. what population was covered, how many people, how many villages within a town/district, etc. Is Magude a town or a province? How big is Magude? Is it more children or adults? Is Magude just someone’s name or Portugese term for something? (I know it is a town because I googled it but you see my point).

- Typically, we call it a pilot if the sample size underpowers the conclusions. I acknowledge that this is a cost-effectiveness study but it would be good to know what was the magnitude of the actual study and its interventions.

Discussion

- Lines 236-242: first paragraph of discussion should more clearly state main outcome

- 246-250 is excellently stated but the last sentence at 250-252 should be rephrased using respectively

- 283-298 should be one paragraph as it is addressing the same issues

- Why do transportation

- Line 312 - should be micro-costing

- Line 318 - you only say sub-Saharan Africa one time so does not need to be acronym

- Lines 309-320 should be a single concluding paragraph that is tighter and shorter with less run on and repetition

Miscellaneous

- I am pretty sure that Regina is an MD in addition to an MPH on your author byline

Reviewer #2: As researchers are in search of the optimal combination of interventions to achieve malaria elimination, in the context of more and more reduced resources, the authors present interesting results contributing to the evidence that a combination of drug-based strategies and intensified control measures could be cost effective. The manuscript is well written and particularly well detailed.

Here are some minor comments for authors’ consideration:

Minor comments

• Line 64: “From an economic point of view, this should have translated into improved efficiency over time, reflected in a lower ICER”.

This sentence sounds leading and could appear as a speculation on the results. I will suggest removing it

• Line 65: The authors refer to the main trial for further details.

For a better understanding of the context of implementation, the authors should consider adding a brief description of the Magude Project.

• Line 67: the authors state that “the package of interventions….. reduced malaria prevalence by 84.7%.” However, it is not mentioned in the manuscript how and when malaria prevalence was measured. The impact measure described in line 132 is a cumulative malaria case incidence.

• Line 80: delete one resource

• Line 82: state the Magude project intervention package here to put in perspective with the counterfactual comparison.

• Methods section:

Related to comment #2: To put the costs and gains in perspective, it would be important for the reader to know:

o The scale of implementation: is Magude a district? This is not mentioned.

o What is the population size?

o What is the malaria endemicity in Magude

• Line 120: The authors mention “intensified IRS -implemented by GoodBye Malaria”.

Is this different from the universal spraying described in lines 92-93 as part of the intervention package? If not, please consider using the same terminology. If it is different, the manuscript should briefly describe how these interventions were implemented in order to help understand the costing aspect and give an idea on how these interventions could potentially be replicated by the programme

• Line 229: The long-term costing does not consider the evidence that the costs of everting a marginal malaria case will increase as malaria transmission declines. As discussed in lines 262-266.

• Line 299: replace “scope” by “scale”

Reviewer #3: Peer Review: Moving towards malaria elimination in southern Mozambique

General comments

This is a very interesting pilot study on cost-effectiveness for malaria mass drug administration as an additional control strategy in southern Mozambique. The findings have implications for other lower-resource settings, where there is interest in exploring novel implementation approaches for moving towards interruption of transmission and possibly also elimination. It is not fully clear to me whether the specific setting for this pilot study fulfills WHO’s recommendation of an elimination setting, and whether this would therefore impact the generalizability of the findings. Close proof-reading of the next draft is necessary – there are some punctuation, grammatical, and other minor errors. Overall, I commend the authors on the project and hope they are able to continue their investigations, including potential alignment of MDA with NTD MDA efforts as another possible implementation approach.

Specific comments

- Lines 32-34: Technically, to achieve elimination, all targets would have to be addressed, since transmission would have to stop across the whole geographic area (and thus all populations). Although I agree that equity is a critical point when thinking about the roll-out of strategies towards achieving elimination.

- Lines 34-35: Not clear why non-excludability and non-rivalry are considered “challenges” or attributes that would create challenges? Surely these are positive attributes, in terms of creating more generalized incentives for elimination?

- Line 37: The regional consideration is critical, also in terms of how reintroduction of cases from neighboring areas that might not achieve elimination would impact overall cost and effectiveness estimations.

- Lines 74-78: It is important to note that the WHO does not generally recommend MDA in areas of moderate to high transmission, which may make it more difficult to justify as a policy in areas “transitioning” to malaria elimination (i.e. implying they are not yet at that stage).

- Line 82: It should be “associated with” not “associated to”.

- Methods: It might be helpful to provide a bit of context about malaria transmission in this setting, to help the reader better understand the timing and approach for the interventions – is malaria transmission seasonal in this context? If yes, was timing to interventions designed specifically with transmission risk in mind (assuming yes but would be helpful to state clearly)? These factors help determine how optimized implementation was of the different interventions, and thus how the cost-effectiveness ratio might change under different circumstances or in other settings.

- Line 101/Figure 1: IRS does not seem to appear in Figure 1 as a strategy implemented by the national control programme (no asterisk next to any of the IRS arrows in the Figure) – is this an oversight, or did the national control programme not deliver any IRS during this time period? If this is the case, line 101 should be revised to reflect this.

- Lines 106-109: Is there any reason to suspect there might have been differences in the effectiveness (or cost) of standard interventions provided by the national malaria programme as compared to those delivered within the context of the Magude project?

- Lines 134-136: Slightly awkwardly worded sentence which makes it difficult to understand. Suggest simplifying and removing excess punctuation.

- Lines 168-171: If additional activities are required though, there could be an argument from the MOH that additional vehicles would be needed (or replacement/maintenance costs would be greater).

- Lines 186-188: It seems strange to assume that malaria incidence would remain the same for the purposes of the model, if interruption of transmission (and eventual elimination) is the end goal?

- Lines 208-209: Does this mean that if incidence in the area decreased (fewer potential cases to avert through MDA), the strategy would become less cost-effective? Does not bode well for using it as an elimination strategy, in that case.

- Lines 210-211: This is a very important point. Do the authors think that it’s an indication that providing resources to support earlier case identification (especially of severe cases) as well as patient referral and appropriate treatment would be more cost-effective than MDA?

- Lines 231-233: But per the Methods, this does not account for reduced incidence of malaria over this time, correct?

- Lines 281-282: This is a useful observation, from a policy standpoint.

- Lines 299-304: The idea of potentially integrating malaria MDA into existing NTD strategies (where mapping studies show potential effective population overlap) is very intriguing – I hope the authors consider this for a future cost-effectiveness study!

Overall, I recommend the manuscript be accepted with minor requested revisions.

6. PLOS authors have the option to publish the peer review history of their article (what does this mean?). If published, this will include your full peer review and any attached files.

Reviewer #1: No

Reviewer #2: No

Reviewer #3: Yes: Claire J. Standley

---

## [Author Response · Author response to Decision Letter 0]

20 May 2020

Response to Reviewers’ comments

Manuscript title: Moving towards malaria elimination in Southern Mozambique: cost and cost-effectiveness of mass drug administration combined with intensified malaria control

Manuscript reference number: PONE-D-20-09618

Requests from the editors 

1. Thank you for including your competing interests’ statement; "The authors have declared that no competing interests exist."

We note that one or more of the authors are employed by a commercial company: ICREA, Pg. Lluís Companys

• Response: ICREA is not a commercial employer. It stands for "Catalan Institution for Research and Advanced Studies". It is a program launched by the Catalan government to fund researchers (https://www.icrea.cat/).

• Response: The same applies as per comment #1: ICREA is not a commercial affiliation. It stands for "Catalan Institution for Research and Advanced Studies". It is a program launched by the Catalan government to fund researchers (https://www.icrea.cat/). 

3. Thank you also for providing the following funding information within your acknowledgements section; "CISM is supported by the Government of Mozambique and the Spanish Agency for International Development (AECID). The Magude project (NCT02914145) was funded by the Bill and Melinda Gates Foundation and Obra Social “la Caixa” Partnership for the Elimination of Malaria in Southern Mozambique (OPP1115265)."

• Response: Thank you . We have removed the funding-related text from the manuscript and suggest updating the Funding Statement as follows (this has also been mentioned and included in the cover letter):

“We acknowledge support from the Spanish Ministry of Science, Innovation and Universities through the “Centro de Excelencia Severo Ochoa 2019-2023” Program (CEX2018-000806-S), and support from the Generalitat de Catalunya through the CERCA Program. CISM is supported by the Government of Mozambique and the Spanish Agency for International Development (AECID). The Magude project (NCT02914145) was funded by the Bill and Melinda Gates Foundation and Obra Social “la Caixa” Partnership for the Elimination of Malaria in Southern Mozambique (OPP1115265). The funders had no role in study design, data collection and analysis, decision to

publish, or preparation of the manuscript.”

Comments from reviewers

REVIEWER # 1

General comments:

This study evaluates the cost-effectiveness of a pilot malaria elimination program in Mozambique called the Magude project. It demonstrates, like other elimination studies, that drastic disease reductions are achieved during active efforts; but unfortunately, these efforts tend to fail in eliminating local transmission and therefore disease burden climbs again once MDA is stopped. Correspondingly, the ICER of the project improved with each MDA round but ultimately did not achieve a highly cost-effective threshold by the end of the project.

Since I am not a health economist expert, but have a great deal of experience with malaria elimination, the methodology of the study design and analysis appear sound and the figures are clear. However, my concerns are two-fold:

1- the overall manuscript is not clearly written or punctuated such that it was difficult to read

• Response: We have revised the manuscript writing in detail and it has been proof-read by other co-authors and an external academic so as to ensure the content is clear and understandable to a broad audience. 

2- the content may be more appropriate to a malaria-specific journal and less so to the readership of PLOS One. Undoubtedly, as with all malaria elimination programs it was a great deal of important work and should be published somewhere. I therefore recommend major revision.

• Response: The main reason why this manuscript was submitted to PLOS One is that it is part of the Magude Project Special PLOS collection. This collection aims to present a comprehensive vision of different components of the malaria elimination strategy implemented in the Magude project, including the impact on epidemiological, entomological and parasitological outcomes observed during the first and second phase of the project as well as the cost-effectiveness of the mix of interventions. In addition, to our knowledge, PloS One has published a substantial amount of malaria-related (and cost-effectiveness) studies and we think that PloS One is actually read by malaria researchers. 

Specific points:

Grammar, punctuation and sentence restructuring are required throughout manuscript but here are some examples:

3- The English is a bit stilted, for example “in the long-term” is not a correct expression and is used multiple times (Lines 24, 31). Long-term is an adjective not a noun. Also elimination is maintained “at zero” not “to zero” (line 24) and lines 43-49 are confusing and not clear. This and the next paragraph have only single sentences in them. There are many run-on sentences in intro that impact clarity. Would recommend a thorough read and edit by a native English speaker.

• Response: We agree with the reviewer. We have worked throughout the document and with the help of different researchers we have proof-read and edit the document in detail. We believe the writing style has significantly improved. Some specific terms -widely used in economics- have been corrected. For example, “the long term” has been changed to “long-term” when used as an adjective -and followed by a noun- and changed to “long term” when “long” is employed as an adjective for “term” (the noun). 

4- Lines 64-67 should be in results or discussion, not intro

• Response: We see the reviewers’ point. While the effectiveness results of the Magude project will be presented in a separate publication (currently under revision at PLosMed), this study aimed to assess its cost-effectiveness. This is why we thought it was worth mentioning the impact findings in the introduction and provide details of those later in the methods section. More specifically, we have included a “study site” sub-section under methods, describing the context of study, the interventions deployed and the main impact figures achieved. 

5- Lines 96-97 need to use different distinguishers for MDA rounds and time frames - too many short hyphens. Figure 1 is nice and clear.

6- Inappropriate use of hyphen versus em dash throughout (or inconsistent - see lines 199-200)

7- Lines 134-135 --- poorly written and punctuated “Based on the observed malaria cases and the estimated cases expected had the intervention not taken place - the counterfactual-, respectively, we estimated resources…”

8- Line 269 - should be em dashes

9- Line 318 - “Mozambique stands among the weakest link countries” - this does not make sense to use stand in this sentence - Mozambique IS one of the weakest links in southern Africa

• Response: We agree with all points above-mentioned (from 5 to 9) and have revised and introduced all suggested changes across the document accordingly as part of the editing process. 

Abstract

10- Last sentence of background should be in results.

• Response: Aligned with our answer to reviewer’s comment #4, the effectiveness results are not part of the scope of this study (these are presented in a manuscript currently under revision at PLOS Med (PMEDICINE-D-20-00604R1)), as this paper aims to present the cost-effectiveness results of the project. 

11- Inappropriate use of short hyphens throughout

• Response: We have revised the correct use of hyphens vs en dashes vs em dashes across the document as part of the editing process.

Introduction/Methods

12- This is called a proof of concept pilot study in abstract but I do not see any Magude Project references or actual descriptions of the study interventions itself aside from Figure 1 and corresponding test. It would be nice to know more to set the stage - e.g. what population was covered, how many people, how many villages within a town/district, etc. Is Magude a town or a province? How big is Magude? Is it more children or adults? Is Magude just someone’s name or Portuguese term for something? (I know it is a town because I googled it but you see my point).

• Response: We agree with the reviewer. We have included a new sub-section under methods (i.e. “Study site”) providing some information to set the stage. In addition, we have included two references (see below) which describe in detail the rationale of the project and the epidemiological and sociodemographic context in the district, as well as details on the interventions and study procedures.

References 

Galatas, B., Nhacolo, A., Marti, H., Munguambe, H., Jamise, E., Guinovart, C., . . . Sacoor, C. (2020). Demographic and health community-based surveys to inform a malaria elimination project in Magude district, southern Mozambique. BMJ Open, 10(5), e033985. doi:10.1136/bmjopen-2019-033985

Galatas, B., Saúte, F., Martí-Soler, H., Montañà, J., Guinovart, C., Munguambe, H., . . . Aide, P. (2020). The Magude project: a before-after study aiming to eliminate malaria in southern Mozambique. Manuscript under review (PLos Med). 

13) Typically, we call it a pilot if the sample size underpowers the conclusions. I acknowledge that this is a cost-effectiveness study but it would be good to know what was the magnitude of the actual study and its interventions.

• Response: We agree with the reviewer and have not included the concept pilot. In addition, we have included a “study site” sub-section under methods, describing the context of study, the interventions deployed and the main impact figures achieved. We have added information on the district population size in that sub-section as well. 

Discussion

14) Lines 236-242: first paragraph of discussion should more clearly state main outcome

• Response: We have re-written the first paragraph of discussion. It now reads: 

(page 16): “This study shows that the economic cost of the Magude project was substantially higher than the routine malaria control activities that would have otherwise taken place in the district. In spite of higher absolute costs, the project was cost-effective by the end of year 3, with an ICER of $987 (CI95% $968–1,006) per DALY averted, a value below the conventional cost-effectiveness threshold of three times the GDP per capita (S1 Fig). We estimate that the project would have still remained cost-effective if achieving at least 65% of the effectiveness (i.e. number of cases averted) observed. This suggests that the mix of interventions delivered through the Magude project would potentially remain cost-effective if implemented in less favourable environmental and socioeconomic contexts”.

15) 246-250 is excellently stated but the last sentence at 250-252 should be rephrased using respectively

16) 283-298 should be one paragraph as it is addressing the same issues

17) Line 312 - should be micro-costing

19) Line 318 - you only say sub-Saharan Africa one time so does not need to be acronym

• Response: We have modified the text according to comments 15-19

20) Lines 309-320 should be a single concluding paragraph that is tighter and shorter with less run on and repetition

• Response: We have modified the last paragraph of the manuscript in the discussion section (page 18): 

 “While other economic questions related to equity, scalability, sustainability and financial affordability associated with moving from control to pre-elimination remain unanswered, this study offers solid evidence on the economic rationale for prioritizing resources on innovative strategies that accelerate the progress towards malaria elimination. The micro-costing approach presented here also provides essential evidence on key inputs for costing extrapolation and scenario development in other settings. Despite the initial high costs and volume of resources associated with its implementation, MDA in combination with existing malaria control interventions appears a potentially cost-effective strategy to accelerate towards malaria elimination in low to moderate transmission settings in SSA.” 

Miscellaneous

21) I am pretty sure that Regina is an MD in addition to an MPH on your author byline

• Response: We have changed accordingly (although later verified the degree abbreviation is not needed according to PLOS ONE formatting requirements. 

REVIEWER #2: 

As researchers are in search of the optimal combination of interventions to achieve malaria elimination, in the context of more and more reduced resources, the authors present interesting results contributing to the evidence that a combination of drug-based strategies and intensified control measures could be cost effective. The manuscript is well written and particularly well detailed.

• Response: We thank the reviewer for appreciating the study. 

Here are some minor comments for authors’ consideration:

Minor comments

1) Line 64: “From an economic point of view, this should have translated into improved efficiency over time, reflected in a lower ICER”. This sentence sounds leading and could appear as a speculation on the results. I will suggest removing it

• Response: We agree with the reviewer. We have removed it. 

2) Line 65: The authors refer to the main trial for further details.

For a better understanding of the context of implementation, the authors should consider adding a brief description of the Magude Project.

• Response: We have removed the sentence from line 65 and provided further details on the project both in the “introduction” (by specifying the interventions included) as well as in a new sub-section included under methods (i.e. “study site”). In this subsection we describe the context of study, the interventions deployed, and the main impact figures achieved. 

3) Line 67: the authors state that “the package of interventions….. reduced malaria prevalence by 84.7%.” However, it is not mentioned in the manuscript how and when malaria prevalence was measured. The impact measure described in line 132 is a cumulative malaria case incidence.

• Response: We have removed this effectiveness figure from the introduction and introduced impact estimates only later under the new sub-section “Study site”. In addition, it is important to bear in mind that this paper will be part of the Magude Project Special PLOS collection. This collection aims to present a comprehensive vision of different components of the malaria elimination strategy implemented in the Magude project, including the impact of the project on epidemiological, entomological and parasitological outcomes. We have also included the reference of the effectiveness manuscript —currently under revision at PLOS Med (PMEDICINE-D-20-00604R1) — were further details can be found on effectiveness outcome measurement and analysis. 

4) Line 80: delete one resource

• Response: Deleted. 

5) Line 82: state the Magude project intervention package here to put in perspective with the counterfactual comparison.

• Response: We have included a description of the intervention package in lines 65-68 Methods section:

6) Related to comment #2: To put the costs and gains in perspective, it would be important for the reader to know:

o The scale of implementation: is Magude a district? This is not mentioned.

o What is the population size?

o What is the malaria endemicity in Magude

• Response: We have included information on the epidemiological and demographic characteristics of the district in the first paragraph of a new subsection under methods (i.e. “study site”) (lines 90-93). We have also referenced a paper providing further sociodemographic details on the context of study.

Reference 

Galatas, B., Nhacolo, A., Marti, H., Munguambe, H., Jamise, E., Guinovart, C., . . . Sacoor, C. (2020). Demographic and health community-based surveys to inform a malaria elimination project in Magude district, southern Mozambique. BMJ Open, 10(5), e033985. doi:10.1136/bmjopen-2019-033985

7) Line 120: The authors mention “intensified IRS -implemented by GoodBye Malaria”. Is this different from the universal spraying described in lines 92-93 as part of the intervention package? If not, please consider using the same terminology. If it is different, the manuscript should briefly describe how these interventions were implemented in order to help understand the costing aspect and give an idea on how these interventions could potentially be replicated by the programme

• Response: Thanks for the comment. We have uniformed terminology and referred to universal IRS (ie. spraying targeted to all households in the district) when referring to the spraying implemented under the Magude project. This is different from the routine IRS that MoH usually does, which is focal IRS targeted to high-burden areas of the district. 

8) Line 229: The long-term costing does not consider the evidence that the costs of everting a marginal malaria case will increase as malaria transmission declines. As discussed in lines 262-266.

• Response: Our scenario assumes that the gains achieved by the third year could be maintained by continuing with a strengthened surveillance and a response system using rfMDA, on top of standard case management, district-wide IRS and LLIN distributions. Given that the approaches to further reduce malaria transmission are not yet clear, it is not possible to provide estimates of the costs of averting marginal malaria cases. Instead, we have only provided indicative figures on the potential cost-savings if health gains can be maintained (without malaria incidence being further reduced but maintained stable) through time. Lines 262-266 are based on findings from other studies and we can only speculate (without direct evidence) on what could be the case in Magude. Importantly, this study is not about the cost-effectiveness of achieving elimination; the study focuses on a phase moving towards elimination.

9) Line 299: replace “scope” by “scale”

• Response: In this manuscript we have differentiated between “economies of scale” (as the number of people treated increase, cost per treatment decreases ) and “economies of scope”(efficiencies formed by variety, not necessarily by volume). In line 299, when we mention the integration with other community-based health interventions implemented by the government, we refer to “economies of scope”. By integrating the delivery of different governmental programs, the unit delivery cost might decrease significantly (shared transport expenses, personnel expenses… and all expenses related to delivering and monitoring the intervention). This would be the case if MDA could be delivered together (integrated) with other door-to-door interventions (i.e. MDA for neglected tropical diseases).

REVIEWER #3: 

General comments

This is a very interesting pilot study on cost-effectiveness for malaria mass drug administration as an additional control strategy in southern Mozambique. The findings have implications for other lower-resource settings, where there is interest in exploring novel implementation approaches for moving towards interruption of transmission and possibly also elimination. 

1) It is not fully clear to me whether the specific setting for this pilot study fulfills WHO’s recommendation of an elimination setting, and whether this would therefore impact the generalizability of the findings. 

• Response: The Magude project was designed to respond to the call for evidence generation in the World Health Organization (WHO) Global Technical Strategy for Malaria 2016–2030 (GTS). The GTS calls for the generation of evidence to identify new tools and strategies to accelerate towards malaria elimination in malaria endemic areas that already have universal access to vector control and case management. Therefore, the project was not aimed to be conducted in a malaria elimination setting as currently defined by WHO, but rather in a representative malaria-endemic setting of Sub-Saharan Africa (SSA) where reducing malaria morbidity and mortality is most pressing. After establishing an enhanced surveillance system and ensuring near-universal access to care and LLINs, the project used district-wide IRS and MDA to drastically reduce transmission in the area. This was successfully achieved and sustained for at least a year, thus demonstrating that rural areas with stable transmission with SSA can feasibly reduce morbidity to pre-elimination settings in a short period of time through the interventions recommended by WHO. 

References

World Health Organization, World Health Organization, Global Malaria Programme. Global technical strategy for malaria, 2016-2030 [Internet]. 2015 [cited 2018 Mar 5]. Available from: http://apps.who.int/iris/bitstream/10665/176712/1/9789241564991_eng.pdf?ua=1

2) Close proof-reading of the next draft is necessary – there are some punctuation, grammatical, and other minor errors. 

• Response: We agree with the reviewer and this issue has been raised by other reviewers as well. We have revised the manuscript writing in detail and it has been proof-read by other co-authors as well as academics out of the field of malaria so as to ensure the content is clear and understandable to a broad audience. 

Overall, I commend the authors on the project and hope they are able to continue their investigations, including potential alignment of MDA with NTD MDA efforts as another possible implementation approach.

• Response: That is indeed very important if MDA for malaria is to be scaled-up in other regions. There are some promising initiatives on integrated mass treatment coverage for neglected tropical diseases in Mozambique that could serve as reference.

Specific comments

3) Lines 32-34: Technically, to achieve elimination, all targets would have to be addressed, since transmission would have to stop across the whole geographic area (and thus all populations). Although I agree that equity is a critical point when thinking about the roll-out of strategies towards achieving elimination.

• Response: We agree with the reviewer. The Magude project impact paper (currently under revision at Plos Med but part of the same PLOS collection as this manuscript) shows a lot of detail on the spatial and individual-level factors associated with LLIN, IRS and MDA coverage, as well as some characteristics of people who were missed during the MDA. Although equity is a much broader concept, these figures provide a magnitude of the coverage and spatial homogeneity of the interventions. We have provided details on attained coverages at the “study site” section and have referenced the impact paper for further details. 

4) Lines 34-35: Not clear why non-excludability and non-rivalry are considered “challenges” or attributes that would create challenges? Surely these are positive attributes, in terms of creating more generalized incentives for elimination?

• Response: Public goods have the attributes of non-excludability (when ‘‘public’’ goods are provided no one can be excluded from their consumption) and non-rivalry in consumption (one person’s consumption does not prevent anyone else’s). Malaria elimination can be considered also a public good, given that no one in a population can be excluded from benefiting from a reduction in risk of infectious disease when its incidence is reduced, and one person benefiting from this reduction in risk does not prevent anyone else from benefiting from it as well them. Although there is significant benefit to be gained from them by many people, there is no commercial incentive for producing them, since enjoyment cannot be made conditional on payment. As there is no global government to ensure that they are produced and paid for, financial cross border mechanisms need to be created and coordinated. Otherwise, this public good (disease elimination) is under-financed. This is well explained in the referenced paper by Richard D. Smith (2003). 

Reference 

Smith, R. D. (2003). Global public goods and health. Bulletin of the World Health Organization, 81(7), 475. 

5) Line 37: The regional consideration is critical, also in terms of how reintroduction of cases from neighboring areas that might not achieve elimination would impact overall cost and effectiveness estimations.

• Response: Indeed, especially for a country such as Mozambique, that borders with countries already transitioning from pre-elimination to elimination of malaria (i.e Eswatini, South Africa). In such contexts, elimination of malaria is only possible through strong cross border collaboration (some regional initiatives have already been created, such as MOSASWA). 

6) Lines 74-78: It is important to note that the WHO does not generally recommend MDA in areas of moderate to high transmission, which may make it more difficult to justify as a policy in areas “transitioning” to malaria elimination (i.e. implying they are not yet at that stage).

• Response: The reviewer is correct given that the latest recommendations from WHO on MDA were based on evidence that was generated before 2015. Since then, the WHO has continued to review the evidence generated by several places in Africa where MDA was evaluated in "high" and "moderate" areas. The results obtained from the Magude project (considered to be a moderate to low area) as well as from other studies, were presented in an Evidence Review Group at WHO, which concluded that MDA could be implemented in moderate transmission areas. However, more evidence was required to make a formal policy recommendation. Therefore, the Magude project, as well as others, aimed to drive this policy change through the generation of operational evidence to inform WHO's policies. As answered in our response to comment #1, the Magude project followed the recommendations proposed by the WHO GTS for Malaria 2016-30, which say that countries or areas with good access to treatment and prevention tools, and a strong surveillance system, should accelerate towards elimination through the use of tools that drastically reduce transmission. The GTS called for evidence on the type of tools that could serve this purpose, among which MDA has been identified as a potential one. Therefore, the Magude project combined existent tools under the premise that the interruption of malaria transmission was biologically plausible when combining MDA with intensified vector control tools. Overall, the study proved that the package of interventions implemented in Magude drastically reduced transmission in a cost-effective way, and these findings may influence WHO and country-based policy. 

7) Line 82: It should be “associated with” not “associated to”.

• Response: Modified.

8) Methods: It might be helpful to provide a bit of context about malaria transmission in this setting, to help the reader better understand the timing and approach for the interventions – is malaria transmission seasonal in this context? If yes, was timing to interventions designed specifically with transmission risk in mind (assuming yes but would be helpful to state clearly)? These factors help determine how optimized implementation was of the different interventions, and thus how the cost-effectiveness ratio might change under different circumstances or in other settings.

• Response: Agree with the reviewer. Thanks for this. We have included a new sub-section under methods called “study site”, where we provide details on the epidemiological context of the district and timing of the interventions. 

9) Line 101/Figure 1: IRS does not seem to appear in Figure 1 as a strategy implemented by the national control programme (no asterisk next to any of the IRS arrows in the Figure) – is this an oversight, or did the national control programme not deliver any IRS during this time period? If this is the case, line 101 should be revised to reflect this.

• Response: Thanks for the observation. The Magude project included univeral IRS campaigns which were conducted with the project partner Good Bye Malaria (achieving a coverage >80%). Given universal IRS happened, routine IRS (focal IRS spraying high-burden areas) did not happen. However, the costs of focal IRS have been computed in our counterfactual scenario, given this is what would have happened in the absence of the Magude project. We have revised the writing and clearly differentiated between universal (implemented by the Magude project) vs focal IRS (routine IRS implemented by MoH) across the document as well as in the tables. 

10) Lines 106-109: Is there any reason to suspect there might have been differences in the effectiveness (or cost) of standard interventions provided by the national malaria programme as compared to those delivered within the context of the Magude project?

• Response: Yes, there are differences. As mentioned in comment #9, apart from the new interventions implemented in the context of the Magude project (i.e. MDA administration), there was an enhancement of the surveillance system, as well as an intensification of vector control tools (i.e. IRS under the Magude project was targeted to all households in the district -universal IRS- while routine IRS deployed by the MoH only targets high burden areas of the district -focal IRS). This translates into different coverage outcomes as well as different costs (see table 1 of the manuscript). The timing of the interventions (done in shorter vs a longer period of time) as well as resources employed for other operational activities (i.e. trainings and supervision visits conducted in better conditions) might have also impacted the quality and effectiveness outcomes associated to the project. 

11) Lines 134-136: Slightly awkwardly worded sentence which makes it difficult to understand. Suggest simplifying and removing excess punctuation.

• Response: We agree and have modified this part accordingly. 

12) Lines 168-171: If additional activities are required though, there could be an argument from the MOH that additional vehicles would be needed (or replacement/maintenance costs would be greater).

• Response: The reviewer is right. In our costing calculations, we do take into consideration the useful life of capital goods (i.e. equipment, transport, infrastructure…) and have imputed maintenance costs. In the case of transport, maintenance costs are directly linked to km travelled. In consequence, an increase in the use of resources (i.e. distances travelled with governmental vehicles) has been reflected in higher maintenance costs. 

13) Lines 186-188: It seems strange to assume that malaria incidence would remain the same for the purposes of the model, if interruption of transmission (and eventual elimination) is the end goal?

• Response: Evidence from the Magude project effectiveness manuscript —currently being under revision at PLOS Med (PMEDICINE-D-20-00604R1)— showed that with an intensive implementation of currently available tools recommended by WHO, large reductions in malaria transmission and burden of disease are achieved. However, if elimination is to be achieved in areas of stable transmission (such as the Magude district), new tools and strategies are required. In consequence, it is not yet clear how to achieve elimination in such areas. Through our long-term scenario we wanted to show that, even if elimination is not achieved, if the gains accomplished by the third year could be maintained by continuing with intensified vector control tools together with a strengthened surveillance and a response system using rfMDA, potential cost-savings would arise in the mid and long term. Importantly, this study is not about the cost-effectiveness of achieving elimination; the study focuses on a strategy to efficiently move towards elimination (pre-elimination phase). Text S1 provides details on the costing scenario assumptions. 

14) Lines 208-209: Does this mean that if incidence in the area decreased (fewer potential cases to avert through MDA), the strategy would become less cost-effective? Does not bode well for using it as an elimination strategy, in that case.

• Response: This means that the strategy proved to be cost-effective and that there was still some margin for the strategy to remain cost-effective even if effectiveness decreased. In our case, even if less cases had been averted than those observed, the intervention would remain cost-effective. But up to what point would the intervention remain cost-effective? It would need to avert at least 24,936 cases (a reduction in effectiveness of 35%, given the observed averted cases were 38,369). It actually may be the case that, as the elimination gets closer, elimination activities no longer become cost-effective as the unit cost of averting a marginal malaria case would largely increase. This is one of the challenges of disease elimination in the short-tem (as discussed in the introduction). Certainly, elimination is cost-effective in the long-term. 

However, this study is not about the cost-effectiveness of achieving elimination (we don’t talk about the costs or cost-effectiveness in the last mile) but rather present the economic rationale of a strategy to efficiently move towards elimination (regardless of not having achieved malaria elimination).

15) Lines 210-211: This is a very important point. Do the authors think that it’s an indication that providing resources to support earlier case identification (especially of severe cases) as well as patient referral and appropriate treatment would be more cost-effective than MDA?

• Response: It is difficult to compare one intervention vs the other. Earlier case identification and management is crucial to control malaria transmission and is considered to be a very cost-effective strategy that any country aiming to transition to pre-elimination stages should have in place. However, additional tools might be needed to further reduce transmission of malaria. This is the case of Magude, where despite having good access to treatment and reasonably effective implementation of vector control and surveillance, further tools (or new combination of already available tools) were needed to further reduce transmission. 

16) Lines 231-233: But per the Methods, this does not account for reduced incidence of malaria over this time, correct?

• Response: Exactly. Following on comment #13, it is not yet clear how elimination can be achieved in areas of stable transmission (such as in the Magude district), and therefore, it is not possible to provide estimates of the costs of averting marginal malaria cases. Instead, we have provided indicative figures on the potential cost-savings if health gains can be maintained across time. As mentioned earlier, this study is not about the cost-effectiveness of achieving elimination but about a strategy to efficiently move towards elimination (pre-elimination phase).

17) Lines 281-282: This is a useful observation, from a policy standpoint.

• Response: Thank you for the appreciation.

18) Lines 299-304: The idea of potentially integrating malaria MDA into existing NTD strategies (where mapping studies show potential effective population overlap) is very intriguing – I hope the authors consider this for a future cost-effectiveness study!

• Response: We agree with the reviewer and this is an issue that has been raised by other reviewers. If the strategy is to be implemented in a feasible and sustainable way by the government, it will need to be integrated with other community-wide public health interventions. Otherwise, the high volume of resources employed for MDA pose serious doubts on the intervention’s scalability. 

Overall, I recommend the manuscript be accepted with minor requested revisions.

---

## [Decision Letter · Decision Letter 1]

19 Jun 2020

Moving towards malaria elimination in Southern Mozambique: cost and cost-effectiveness of mass drug administration combined with intensified malaria control

PONE-D-20-09618R1

Dear Dr. Crivillé,

We’re pleased to inform you that your manuscript has been judged scientifically suitable for publication and will be formally accepted for publication once it meets all outstanding technical requirements.

Kind regards,

Luzia Helena Carvalho, Ph.D.

Academic Editor

PLOS ONE

Additional Editor Comments (optional):

Reviewers' comments:

Reviewer's Responses to Questions

**Comments to the Author**

1. If the authors have adequately addressed your comments raised in a previous round of review and you feel that this manuscript is now acceptable for publication, you may indicate that here to bypass the “Comments to the Author” section, enter your conflict of interest statement in the “Confidential to Editor” section, and submit your "Accept" recommendation.

Reviewer #2: All comments have been addressed

2. Is the manuscript technically sound, and do the data support the conclusions?

Reviewer #2: Yes

3. Has the statistical analysis been performed appropriately and rigorously? 

Reviewer #2: Yes

4. Have the authors made all data underlying the findings in their manuscript fully available?

Reviewer #2: Yes

5. Is the manuscript presented in an intelligible fashion and written in standard English?

Reviewer #2: Yes

6. Review Comments to the Author

Reviewer #2: (No Response)

7. PLOS authors have the option to publish the peer review history of their article (what does this mean?). If published, this will include your full peer review and any attached files.

Reviewer #2: No

---

## [Editor Report · Acceptance letter]

25 Jun 2020

PONE-D-20-09618R1 

Moving towards malaria elimination in southern Mozambique: cost and cost-effectiveness of mass drug administration combined with intensified malaria control 

Dear Dr. Cirera:

I'm pleased to inform you that your manuscript has been deemed suitable for publication in PLOS ONE. Congratulations! Your manuscript is now with our production department. 

Kind regards, 

on behalf of

Dr. Luzia Helena Carvalho 

Academic Editor

PLOS ONE